# Scoping Review—The Association between Asthma and Environmental Chemicals

**DOI:** 10.3390/ijerph18031323

**Published:** 2021-02-01

**Authors:** Tiina Mattila, Tiina Santonen, Helle Raun Andersen, Andromachi Katsonouri, Tamás Szigeti, Maria Uhl, Wojciech Wąsowicz, Rosa Lange, Beatrice Bocca, Flavia Ruggieri, Marike Kolossa-Gehring, Denis A. Sarigiannis, Hanna Tolonen

**Affiliations:** 1Finnish Institute for Health and Welfare, PO Box 30, 00271 Helsinki, Finland; hanna.tolonen@thl.fi; 2Department of Pulmonary Diseases, Heart and Lung Center, Helsinki University Hospital and Helsinki University, Meilahti Triangle Hospital, 6th Floor, PO Box 372, 00029 Helsinki, Finland; 3Finnish Institute of Occupational Health, PO Box 40, 00032 Helsinki, Finland; tiina.santonen@ttl.fi; 4Environmental Medicine, Department of Public Health, University of Southern Denmark, DK-5000 Odense, Denmark; hrandersen@health.sdu.dk; 5State General Laboratory, Ministry of Health, PO Box 28648, 2081 Nicosia, Cyprus; akatsonouri@sgl.moh.gov.cy; 6National Public Health Center, 1097 Budapest, Hungary; szigeti.tamas@nnk.gov.hu; 7Environment Agency, 1090 Vienna, Austria; maria.uhl@umweltbundesamt.at; 8Nofer Institute of Occupational Medicine, 91-348 Lodz, Poland; wojciech.wasowicz@imp.lodz.pl; 9German Environment Agency, 14195 Berlin, Germany; rosa.lange@uba.de (R.L.); marike.kolossa@uba.de (M.K.-G.); 10Department of Environment and Health, Istituto Superiore di Sanità, 00161 Rome, Italy; beatrice.bocca@iss.it (B.B.); flavia.ruggieri@iss.it (F.R.); 11Technologies Division, Environmental Engineering Laboratory Department of Chemical Engineering and HERACLES Research Center on the Exposome and Health Center for Interdisciplinary Research and Innovation, Aristotle University, GR-54124 Thessaloniki, Greece; sarigiannis@auth.gr

**Keywords:** asthma, environmental chemicals, exposure, HBM4EU, occupation

## Abstract

Asthma is one of the most common chronic diseases worldwide affecting all age groups from children to the elderly. In addition to other factors such as smoking, air pollution and atopy, some environmental chemicals are shown or suspected to increase the risk of asthma, exacerbate asthma symptoms and cause other respiratory symptoms. In this scoping review, we report environmental chemicals, prioritized for investigation in the European Human Biomonitoring Initiative (HBM4EU), which are associated or possibly associated with asthma. The substance groups considered to cause asthma through specific sensitization include: diisocyanates, hexavalent chromium Cr(VI) and possibly p-phenylenediamine (p-PDA). In epidemiological studies, polyaromatic hydrocarbons (PAHs) and organophosphate insecticides are associated with asthma, and phthalates, per- and polyfluoroalkyl substances (PFASs), pyrethroid insecticides, mercury, cadmium, arsenic and lead are only potentially associated with asthma. As a conclusion, exposure to PAHs and some pesticides are associated with increased risk of asthma. Diisocyanates and Cr(VI) cause asthma with specific sensitization. For many environmental chemicals, current studies have provided contradicting results in relation to increased risk of asthma. Therefore, more research about exposure to environmental chemicals and risk of asthma is needed.

## 1. Introduction

Asthma is one of the most common chronic diseases with an estimated global prevalence of 16% appearing in all age groups. Characterization of asthma includes airway inflammation, variable airway obstruction and heterogeneous symptoms [1,2,3]. Asthma causes a high burden and economic costs for society and individuals throughout hospitalizations, disability, premature deaths and medications. Though the global prevalence of asthma is increasing, disability-adjusted life years (DALY) and mortality have decreased [1,2,3,4]. Mechanisms lying behind asthma’s development are complex and include host factors such as genetics and sex, and environmental factors, like exposure to allergens or smoking [1,3]. Additionally, various environmental chemicals may affect the risk of asthma development and can escalate asthma symptoms [1,3,5,6,7,8,9,10,11,12,13,14,15,16,17,18,19,20,21,22,23,24].

Showing an epidemiological association between environmental chemical exposures and chronic disease is difficult. Researchers often study health effects of one environmental chemical at a time, yet less is known about the health effects of real-life exposure to mixtures. Humans are exposed simultaneously to small concentrations of many hazardous environmental substances in addition to multiple other risk factors for diseases. For instance, additional to predisposing environmental factors such as to tobacco smoke and other air pollutants such factors as genotype, childhood infections and being overweight increase the probability for asthma, which is not one single disease but consists of various “phenotypes”. As another example, air pollution (a mixture of chemical and biological pollutants) is associated with asthma and allergic diseases in cellular and animal models, yet no epidemiological causal relationship between individual air pollutants and asthma is established [1,3,5,7,25,26,27].

An epidemiological association should be distinguished from the determined causal association. For asthma and chemicals, this distinction is characterized as specific sensitization leading to hypersensitivity of the airways following inhalation of the substance, which may involve the induction of specific immunoglobulin E (IgE) for the substance or the diagnosis is based on the evidence of the positive reactions in bronchial challenge tests (occupational exposure, low molecular weight chemicals) [28].

The European Human Biomonitoring Initiative (HBM4EU) is a joint effort of 30 countries (26 European Member States, three associated countries and Switzerland) and the European Environment Agency co-funded by the European Commission. The main objective is to assess human exposure to chemicals and to generate knowledge on chemical exposures and their health effects at the population level by means of human biomonitoring [25]. A set of substances was chosen for investigation and development of chemical policy in the frame of HBM4EU, using prioritization criteria, such as an open policy relevant question to support current EU policy making, established concerns about human health effects and evidence of human exposure at the EU level [25].

This scoping review focuses on HBM4EU priority substance groups with known or possible associations with asthma and describes potential sources of exposure and how human exposure to these chemicals can be measured.

## 2. Methods

Scoping documents have been developed for all HBM4EU-prioritized substance groups and are publicly available on the project’s website. The focus of HBM4EU is to support European and national chemical policies by building knowledge on exposure to selected chemicals and related health impacts for European populations. The substances under investigation were selected by a comprehensive, international group of regulators, researchers and other substance experts in a structured prioritization process, based on the need for knowledge on human exposure to these substances in Europe and possible health effects [25]. A literature search was performed in PubMed using the terms “asthma” and each of the HBM4EU prioritized chemicals independently, so as to identify those substances with a reported association or a possible association with asthma. The most significant findings from the resulting original studies and review publications of recent years are summarized herein (from 1994 until 2020). This work is a scoping review of the literature and not a systematic review, since it is not based on a fully comprehensive literature search [29].

## 3. Results

### 3.1. Chemicals Associated with an Increased Risk of Asthma

HBM4EU-priority chemicals associated with asthma are diisocyanates, hexavalent chromium (Cr(VI)) (both cause specific sensitization), polyaromatic hydrocarbons (PAHs) and organophosphate insecticides [8,12,18,23,24]. The following chemicals are possibly associated with asthma: phthalates, per- and polyfluoroalkyl substances (PFASs), pyrethroid insecticides, mercury, cadmium, arsenic, lead and p-Phenylenediamine (p-PDA, a potential specific sensitizer) [10,11,12,13,14,15,16,17,19,20,21,22].

#### 3.1.1. PAHs

PAHs are a group of lipophilic, semi-volatile compounds with 2 to 7 aromatic benzene rings generated when organic materials are combusted incompletely (burning in high temperatures with low oxygen). PAHs are ubiquitous, widespread and released all over from both natural and anthropogenic sources and transported over long distances before precipitation into soils, sediment and vegetation. PAHs can react with ultraviolet light and other pollutants (e.g., ozone, nitrogen oxides and nitrate radicals) leading to the formation of PAH derivatives (e.g., nitrated and oxygenated PAHs). Factors such as temperature and humidity affect reactions. Humans are exposed directly to PAHs and PAH derivatives or through bioaccumulation in the food chain [7,18,27]. PAHs are known or suspected to be mutagenic, carcinogenic and teratogenic, yet, toxicity between PAHs and their derivatives varies and depends on concentration, time and route of the exposure [7,25,30]. There are legislation and regulations for PAHs in the EU and separate countries [7,25].

PAHs associated with fine particles enter the lungs causing inflammation and affecting respiratory health. According to epidemiological studies, there is an association between exposure to PAHs, concentrations of air pollutants and the development of allergic and non-allergic asthma, increased symptoms of asthma, risk of asthma exacerbations and lung function decrease, yet according to current data there is a low level of evidence. The strongest evidence is shown for the association between development of asthma and lung function in children [18,27,30,31]. Various mechanisms are likely to play a role in these pathophysiological processes including inflammation, immunoglobulin E (IgE), mast cells’ and eosinophils’ mediated reactions, oxidative stress, and epithelial and endothelial dysfunction [18,27,32,33]. The risk of asthma is associated with the duration and dose of exposure to PAHs. However, normally, PAHs are one of many air pollutants, and therefore it is difficult to study the effects of each single chemical (mixed exposure) [27,30,31].

#### 3.1.2. Diisocyanates

Diisocyanates are considered as a part of the aniline family used in different industrial applications such as manufacturing polyurethanes and hardeners in various products. Mostly, used diisocyanates include methylene diphenyl diisocyanate (MDI), toluene diisocyanate (TDI) and non-aromatic hexamethylene diisocyanate (HDI). Diisocyanates are produced with an annual volume of 2.5 million tons in the EU and they are one of the most common causes of occupational asthma in Europe. To reduce the number of occupational diseases caused by diisocyanates, the EU is planning to restrict the use of diisocyanates and to set an EU-wide occupational limit value for exposure to diisocyanates in workplaces [25,27,34,35,36].

Diisocyanates cause specific sensitization, which is associated with respiratory symptoms. Moreover, skin sensitization may occur after exposure to diisocyanates. Even a very low exposure to diisocyanates may cause sensitization and asthma. Asthma induced by isocyanates is sometimes IgE-mediated, however, often specific sensitization is observed in specific bronchial challenge tests without specific IgE. Diisocyanates may also cause asthma by irritating mechanisms (reactive airways dysfunction syndrome (RADS) [8,25,35,37].

#### 3.1.3. Cr(VI)

Chromium exists in oxidation states from −2 to +6, and of those, Cr(III) and Cr(VI) are the most frequently found in the environment. Cr(VI) is mostly a manmade oxidizing compound which is mobile in nature. Contamination for Cr(VI) occurs mainly in large industrial emissions (metallurgical and chemical industries). There is a 3–4.5% annual growth in the demand of Cr in the EU [25,38]. Cr(VI) is carcinogenic, genotoxic and causes various health effects such as skin irritation, nasal epithelium damage and lung fibrosis [25,38]. There are legislation and regulations to limit the exposure to Cr(VI) from occupational sources and from different consumer articles [25,38].

Due to the widespread use of Cr(VI) in various industrial sectors, workers can thus be considered the main vulnerable population in developing asthma. Numerous case studies revealed a link between occupational asthma and inhalation of certain particulate Cr(VI) salts/oxides mainly in electroplaters, stainless steel welders, surface treatment workers and construction workers [24,39,40,41,42]. Underlying mechanisms involved in occupational asthma caused by Cr(VI) are not fully elucidated, but may include non-immunologic and immunologic mechanisms, and the latter can be IgE-dependent or non-IgE dependent [39]. A positive association between asthma in adults and urinary Cr was observed in China [12].

#### 3.1.4. Pesticides

Pesticides are a large group of chemicals with various chemical structures and from multiple substance groups used for controlling pests, such as insects and fungus [25,43]. Several pesticides are known, or suspected to be, neurotoxic, carcinogenic and/or endocrine disruptors. The use of various pesticides has varied by time and by country. The EU regulates the use of pesticides, for instance by limiting the maximal residue concentrations in food [23,25].

Organophosphate and pyrethroid insecticides are two major classes of pesticides included in the HBM4EU-project. Organophosphates are a group of chemicals of which chlorpyrifos and dimethoate are the most important ones. They are used in agriculture as insecticides and acaricides. They inhibit acetylcholinesterase activity and possess high acute toxicity, which includes, for instance, vomiting, diarrhea, abdominal cramps, dizziness, eye pain, blurred vision, confusion, paralysis, bronchoconstriction, in severe cases respiratory failure and even death (“cholinergic syndrome”) [23,25,43]. Pyrethroids are in the EU and are globally one of the major classes of insecticides which have partly replaced organophosphates. Pyrethroids are used, for instance, in plant protection, wood preservation and to combat insects in buildings and animal facilities. Chemically, they are synthetic analogs of pyrethrins, and they depolarize axonal sodium and other ion channels and have partly replaced organophosphates. An acute high exposure to pyrethroids causes instances of dyspnoea, coughing, bronchospasm, nausea, vomiting, dermal effects and peripheral neural symptoms [23,25,43].

Multiple epidemiological studies have shown associations between pesticides and asthma. The strongest association was seen for organophosphate insecticides. Use of insecticides was also associated with exacerbation of asthma among subjects with allergies. Most of the evidence comes from studies among occupationally exposed farmers and pesticide applicators, while there are fewer studies with conflicting results in the general population [23,44,45,46]. A large Canadian cohort study found an association between exposure to organophosphates and reduced lung function in the adult general population and pyrethroids and reduced lung function in children and adolescents [45,47].

#### 3.1.5. Phthalates and Hexamoll^®^ DINCH

Phthalates are a group of substances with a chemical structure being esters of phthalic acid of which ortho-phthalates can be divided into high molecular weight (HMW) and low molecular weight (LMW) phthalates depending on their molecular structure. They are used in millions of tons per year globally. Most phthalates are used primarily as plasticisers as they increase the flexibility and service life of soft poly vinyl chloride (soft-PVC). Phthalates are not chemically bound to the materials and can leach, migrate or evaporate into indoor air, the atmosphere, foodstuffs or other materials. Some phthalates are known toxicants to male reproductive development and disturb the endocrine system. They can act in a dose-additive manner and thereby cause a cumulative risk for health [6,20,21,48,49]. Some, but not all, are restricted in Europe but not necessarily in other non-EU countries, and therefore, exposure to imported and to recycled old plastic products also containing restricted phthalates continues in the EU [48].

Epidemiological studies have showed a possible association between exposure to phthalates and asthma. Phthalates might exacerbate already-existing respiratory symptoms [6,20,21]. For instance, in children, exposure to phthalates at home is associated with asthma and allergies. A survey from the U.S. found that HMW phthalates may cause allergy, allergic symptoms and sensitization in adults but not in children, yet in the same survey, phthalates were associated with self-reported asthma in children [50,51]. In adults, exposure to heated PVC fumes is possibly associated with asthma. Experimental studies in vitro and in mice have reported that several phthalates may modulate the murine immune response to co-allergens and act as adjuvants in atopy and allergic reactions [6,20,21,52].

#### 3.1.6. PFASs

PFASs are a large group of highly stable carbon fluorine compounds of which many are very persistent and bio-accumulative; they are widely used in major industry sectors and consumer applications. PFASs are ubiquitous contaminants and partly highly mobile in the environment; they are found in biotic and abiotic environments and food. Of PFASs, specifically PFOA and PFOS are well studied and cause harm for development, are toxic for some target organs and are suspected to be carcinogenic. In humans, the longest half-life for PFASs is up to 8.5 years, with an elimination range from 10 to 56 years [9,25,53,54,55,56]. Due to the overall concerns for human health and the environment, elements of an EU-strategy for PFASs have been worked out and a restriction proposal for PFASs for non-essential uses is currently under preparation [25,53]. Moreover, the recently published chemicals strategy for sustainability towards a toxic-free environment intends to phase out the use of PFASs in the EU, unless their use is essential [57].

There are a limited number of studies on children, adolescents and adults with no consistent results about the association between PFASs and asthma. Some studies on children and adolescents report no association between PFASs and asthma [9,26,58], allergies and IgE levels [9,26]. Two studies reported a possible association [9,59] and a third that some PFASs were associated with increased risk of self-reported asthma but not with current asthma or a wheeze [60]. In a study from Taiwan, serum concentrations of PFASs were associated with asthma, markers and severity for asthma [22], and another study from Norway showed a positive association of exposure to PFAS and clinically severe asthma in adolescents [60]. The scientific opinion of the European Food Safety Authority (EFSA) CONTAM-panel summarises that there are no or inconsistent associations with asthma and allergies for prenatal and postnatal PFAS exposures. This is in line with two recent reviews on health effects [61] and specifically asthma related outcomes [62] of PFAS. However, the risk characterisation of EFSA is based on the effects on the immune system in humans at very low exposure concentrations, which is also supported by the effects in laboratory animals [53]. Ongoing work in HBM4EU shall provide further insight into PFAS-related effects on the immune system including asthma and allergies.

#### 3.1.7. p-PDA

Aniline and its derivatives are aromatic amines which are known or suspected to be carcinogenic and genotoxic [25]. The derivative p-PDA is a common contact allergen causing skin sensitization (contact dermatitis) and exposure is possibly associated with increased risk of occupational asthma and rhinitis through sensitization [19,25,34,63]. Some anilines and their derivatives are restricted in the EU and under the REACH regulations [25,63].

#### 3.1.8. Mercury

Mercury (Hg) is a highly toxic heavy metal which bio-accumulates in food chains and remains in circulation for thousands of years when released. There are natural and anthropogenic sources for Hg which travel over long distances in the air. Mercury exists in various forms, of which exposure sources, target organs, toxicity and metabolism differ [25,38,64]. Organic methylmercury compounds cause the highest risk to human health because they accumulate in the food chain—mainly in large predatory fish—and cause severe and irreversible effects on the central nervous system, even at very low levels [25,38,64]. The use and emission of Hg is restricted with several regulations in the EU and globally [25,38].

There is no definite association between asthma and exposure to Hg. In children, the blood concentration of Hg was shown to both, associate [11] and not to associate with asthma [10,65]. The body burden of Hg might be associated with acute atopic eczema and total IgE measured in children [66].

#### 3.1.9. Cadmium

Cadmium (Cd) is a heavy metal with cumulative toxic effects. The yearly global consumption is 22,000 metric tons. Cd is released from natural sources, for instance in volcanic eruptions and forest fires, and anthropogenic sources, for instance corrosion protection of steel and soldering metal in alloys [25,38]. Cd is carcinogenic and its main target organs are kidneys and bone. The level of exposure affects toxicity, yet only moderate environmental exposure might increase the risk of osteoporosis and age-related decrease in kidney function. There are restrictions in the EU and globally on the use of cadmium [25,38].

Exposure to inhaled Cd might be associated with asthma, however smoking confounds this association while cigarettes include cadmium and smoking is associated with asthma. In smokers, the blood concentration of Cd was associated with wheeze and asthma and in all subjects with lower forced expiratory volume in 1 s (FEV_1_) per forced vital capacity (FVC) and fractional exhaled nitric oxide (FeNO) [15] and urinary concentration of Cd with asthma [12]. The blood level of Cd had an association with self-reported asthma [65].

#### 3.1.10. Arsenic

Arsenic (As) is a chemical element and a significant global environmental toxicant released from natural and anthropogenic sources, appearing everywhere in our environment and circulating in water, air and living organisms and concentrating in soil for years. There are multiple forms of As of which inorganic arsenic (iAs) combined with other elements, such oxygen and sulphur, is the most toxic [13,25,38]. The major toxicity of iAs originates from the natural geological sources such as contaminated drinking water. iAs is for instance carcinogenic and causes vascular diseases and neurotoxicity. Various compounds have various physicochemical properties and toxicity and vary between individual metabolisms related to age, gender and life stage such as pregnancy [13,25,38]. There are international regulations for the use of As [25,38].

There are a limited number of studies on the association between exposure to iAs and its methylated forms and asthma and, therefore the association is undetermined. Chronic exposure to As was associated with various respiratory symptoms, decrease in lung functions, dyspnoea, asthma and increased IgE [13,14], and respiratory symptoms in children in China [41]. A recent study clearly indicates that chronic exposure to As in Bangladesh is associated with characteristic features of asthma [67].

#### 3.1.11. Lead

Lead (Pb) is a toxic heavy metal found in air, water and soil, especially in urban and industrial areas. Lead is used up to 10,000,000 tons per year in the EU and the human exposure to Pb occurs mainly from anthropogenic sources. Pb is present in different inorganic and organic forms which have various properties [25,38]. Inorganic lead compounds are neurotoxic. Foetuses and young children are especially vulnerable because their nervous system is particularly sensitive to the effects of Pb [25,38]. The EU and separate countries have introduced different regulations on Pb [25].

There are quite a few studies investigating the association between exposure to lead and asthma, and therefore, the association is undetermined. Increased Pb blood levels are associated with bronchial hyperresponsiveness, total IgE and decrease in FEV_1_ and FVC [16,17,25,68]. In a recent study the blood Pb level was associated with self-reported asthma [65].

### 3.2. Exposure Routes

Table 1 summarizes known exposure routes and common sources for the investigated environmental chemicals. For many environmental chemicals, drinking water, food and air are known exposure sources, together with cosmetics, clothing and materials used in our living environment.

### 3.3. Asthma and Occupational Exposure to Environmental Chemicals Included in HBM4EU

Occupational exposure occurs for all chemicals included in this article, yet an association with asthma has only been established for a few chemicals. Prevalence of asthma in workers exposed to diisocyanates without appropriate protections ranges from 5 to 15%. Asthma often persists after removal of the exposure. Groups having a specific risk of exposure include those producing and working with diisocyanates [8,25,35,36,37]. Groups at risk of occupational asthma caused by Cr(VI) include, for example, stainless steel welders and surface treatment workers using Cr(VI) [24,25,38,39,69]. Hairdressers working with p-PDA and related compounds might have an increased risk of developing occupational asthma [19,63].

Persons producing and using pesticides have shown an epidemiological association between occupational exposure and asthma [23,25,43,44]. Occupational exposure to PAHs may occur in many occupations, for instance in foundries, coke oven workers, wood preservation and asphalt workers. However, there is no proven causal association between occupational exposure to PAHs and asthma. As an exception, occupational external tobacco smoke might be associated with onset of adult asthma [7,18,27,30].

There are no data to support associations between occupational exposure to phthalates, PFASs, mercury, cadmium, arsenic or lead and asthma [34].

### 3.4. Human Biomonitoring of Environmental Substances

Table 2 summarizes the matrices in which the internal exposure of humans to environmental substances can be determined and additional markers which should be taken into consideration. All examined substances can be measured either in urine or blood. For PAHs, diisocyanates, Cr(VI), pesticides, phthalates, cadmium and arsenic urine are the most commonly used matrices, while PFASs, p-PDA, mercury and lead are usually measured in blood.

### 3.5. Sensitive Population Groups and Association with Asthma

Table 3 summarizes specifically sensitive population groups for each chemical and whether these have an increased risk of asthma. Generally, those who are exposed the most have an increased risk, and foetuses and children are the most vulnerable for adverse effects of chemicals, because the organs are under development.

## 4. Discussion

Exposure to diisocyanates and Cr(VI) causes asthma through sensitization. PAHs and some pesticides have an epidemiological association with asthma. This article focused only on chemicals included in HBM4EU and introduced briefly their association with asthma. There are probably multiple other chemicals associated with asthma, though not included here, and other which are not studied for the association with asthma.

Due to many other toxic properties of these chemicals, there are regulations and restrictions for their use [25]. The use of PAHs, diisocyanates, Cr(VI), pesticides, mercury, cadmium, arsenic and lead is regulated and diisocyanates, phthalates, PFASs and anilines regulated or restricted globally, in the EU or in separate countries in the EU [7,9,23,25,27,34,35,38,48].

Multiple factors make it difficult to draw an exact conclusion from the association between environmental chemicals and asthma. Mostly there were limited number of studies with varying quality and results and many studies were performed in Asia where the air is often polluted by multiple chemicals. Most studies analysed the effects of a single chemical, yet, in real life subjects are exposed to mixtures of various chemicals [3,5,23,25].

## 5. Conclusions

Some environmental chemicals have been shown to be associated with asthma, various asthma phenotypes, and to exacerbate symptoms of asthma in occupationally exposed population groups or in the general population. Some cause asthma through specific respiratory sensitization and for the others an association has been shown in epidemiological studies. Data remain still scarce, and even less is known about the health effects related to combined exposure to various chemicals. These research gaps need to be addressed in future studies on the health effects of environmental chemicals to support policy making for public health and environmental protection.

## Figures and Tables

**Table 1 ijerph-18-01323-t001:** Exposure routes and some of the most common sources of exposure.

Substance	Route of Exposure	Most Common Source of Exposure
PAHs [7,18,25,27,30,64]	Ingestion, inhalation, dermal contact	Urban areas: fuel combustion, smoking, municipal waste, open burning inside, building materials and furniture Industry: e.g., manufactures with smelting processes, conversing wood pulp to paper, burning coal Nature: volcanic activities, seepage of petroleum and coal, humus and plant leave, roots Drinking water, food: grilling, roasting, frying, meat, fish, milk, contaminated cereals, vegetables
Diisocyanates [8,25,35,36,37]	Inhalation, dermal contact	Foams, sealants, coatings, plastic, medicinal products made, for instance, from polyurethane
Cr(VI) [25,38,69]	Inhalation, ingestion, dermal contact	Food, drinking water, smoking cigarettes, pigments and dyes, cosmetics, orthopaedic implants, leather articles
Pesticides [23,25,43]	Ingestion, inhalation, dermal contact	Direct exposure related to pesticide manufacturing or use Drinking water, food: fruits, vegetables, cereals, lentils, linseeds, soya beans, dry peas, tea, dust (ingested by children)
Phthalates (substance groups and substitute) [6,20,21,47,48,64]	Ingestion, dermal contact, inhalation, intravenously	
(1) High molecularweight (HMW) ortho-phthalate in soft-PVC		Fatty food, flooring, wires, cables, sport equipment, toys, coated textiles, footwear, synthetic leather, medical devices
(2) Low molecular weight (LMW) in soft-PVC and other applications		Gelling plasticizers, paints, dispersions, adhesives, cosmetics, solvent in insect repellents
(3) Hexamoll^®^ DINCH		Soft PVC medical devices: blood bags, food contact materials, sports equipment, textile coatings, wallpapers, paints, inks, adhesives, cosmetics and toys
PFASs [9,25,53,54]	Ingestion, inhalation	Contaminants in food and drinking water from production and use: ingredients of surfactants and surface protectors, cooking ware, other food contact materials, cosmetics, lubricants, pharmaceuticals, printing, fire fighting foams
p-PDA [19,25,63]	Dermal contact, inhalation	Cosmetics, hair dyes, tattoo inks
Mercury [25,38,64]	Inhalation, ingestion, dermal contact, injection	Food: fish and seafood Dental amalgams, electrical, medical and laboratory equipment, batteries, pigments, cosmetics (skin creams), incineration, combustion of fossil fuels
Cadmium [25,38,64]	Inhalation, ingestion	Smoking, drinking water Food: seafood, liver, kidney, wild mushrooms, flaxseed, cocoa powder, cereals, potatoes and vegetables (grown on contaminated soil)
Arsenic [13,25,38]	Ingestion, inhalation, dermal contact	Drinking water, food (rice, plant derived food, seafood) Semiconductor and other electronic devices, pesticides, herbicides, insecticides, cotton desiccants, paints
Lead [25,38,64]	Inhalation, ingestion, transplacental	Drinking water through systems with lead solder and lead pipes, atmospheric particles, food, lead-based paints and pigments, lead-containing building materials, furniture, leather products, e-waste, herbal and traditional medicines, cosmetics, toys

**Table 2 ijerph-18-01323-t002:** Matrices commonly used for measuring exposure to substances and factors to consider when measuring.

Substance	Measured in	Remarks
PAHs [7,18,30,64]	**Urine**^1^, other body fluids	Possible to measure several PAHs or their metabolites. Most commonly used biomarkers 1-hydroxypyrene and 1- and 2-napthols. Half-life 5 h–17 d
Diisocyanate [8,25,35]	**Urine**^1^, skin tests (Prick), blood tests (adducts, IgE, IgG), serial peak expiratory flow (PEF) measurement at work place, specific inhalation challenge to diisocyanates	Urinary diamines, haemoglobin and albumin adducts and IgE used to measure exposure, other sensitization
Cr(VI) [25,38,69]	**Urine**^1^, whole blood, plasma or red blood cells	Measurement of urinary, whole blood or plasma chromium levels is not specific for Cr(VI) exposure since also exposure to Cr(III) affects the levels. Only red blood cell chromium can be considered a specific biomarker for Cr(VI)
Pesticides [25,43,64]	**Urine** ^1^	Variable group of compounds, biomonitoring possibilities vary
Phthalates and metabolites [6,48,64]	**Urine**^1^, blood, saliva and breast milk	Non-persistent and have a short half-life in the body, therefore the levels of phthalate metabolites show a high daily variation
PFASs [9,25,53]	**Blood (serum)**^1^, breast milk, urine	Ubiquitous and persistent pollutants with a long half-time in blood Multiple different substances, biomonitoring methods not available for all
p-PDA [19]	**Blood (IgE)**^1^, patch test, lung function testing, inhalation challenge test	Measures sensitization, no validated methods available for the biomonitoring of exposure although some published reports on the measurement of its metabolites in urine or blood available
Mercury [25,38,64]	**Blood**^1^, scalp hair, urine	Different states have different kinetic properties Half-life in blood 1–3 weeks for inorganic and elementary mercury and 50 days for methylmercury
Cadmium [25,38,64]	**Urine**^1^, blood, placenta (exposure in pregnancy), faeces	Urinary analysis of cadmium levels reflect long-term accumulation Individual factors (sex, age, diet, smoking, metabolism etc) influence the concentration of cadmium in urine Half-life varies depending on organ/matrix, for instance 10–40 years in kidneys
Arsenic [25,38]	**Urine**^1^, blood, hair	40–60% eliminated through urine, different forms of arsenic (MMA, DMA, As5+, As3+) can be measured in urine.
Lead [25,38,64]	**Blood**^1^, long term exposure: bone, teeth, hair, nail	The half-life generally long but varies between different organs, for instance in adult the half-life in blood approximately 1–2 months and in bones 10–30 years

^1^ The most important matrix bolded.

**Table 3 ijerph-18-01323-t003:** Sensitive population group for each chemical and shown association with asthma in this population group.

Substance	Sensitive Population	Association with Asthma
PAHs [18,27,30]	Children, subjects with allergies	Yes
Diisocyanates [8,34,35,36,37]	Subjects producing and working with diisocyanates	Yes
Cr(VI) [25,38]	Subjects occupationally exposed to Cr(VI)	Yes
Pesticides [25,43]	Pregnant women, children early in life and in postnatal period, foetuses	Yes
Phthalates and Hexamoll^®^ DINCH [21,25]	Children early in life and in prenatal period	No
PFASs [9,53,58,60]	Pregnant women, children (especially early in life), foetuses	Possibly
p-PDA [19,34]	Subjects occupationally exposed to p-PDA	Possibly
Mercury [25,38,64]	Foetuses, children in postnatal period and early in life, heavy seafood consumers	No
Cadmium [25,38]	Pregnant and postmenopausal women, elderly, children in postnatal period and toddlers	No
Arsenic [25,38]	Children	No
Lead [25,38]	Foetuses and children	No

## Data Availability

All data used and referred can be found in PubMed or other sources according to reference list.

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
