# Peer review of "Scoping Review—The Association between Asthma and Environmental Chemicals"

_ijerph, 2021, doi:10.3390/ijerph18031323_

Round 1
Reviewer 1 Report
Dear authors,
I revised your replies and your updated manuscript. I consider that it has improved with the information and references added.
I believe that it has improved, and I do not have further suggestions for edits.
Thank you.
This manuscript is a resubmission of an earlier submission. The following is a list of the peer review reports and author responses from that submission.
Round 1
Reviewer 1 Report
The review has scientific relevance, however I suggest enriching the study by adding some information such as:
- Search period in the database;
- Maintain a standard of information about each pollutant (chemical structure, sources, transport, reactions with other pollutants, legislation, concentration limit in Europe which can be a range, health effects, ...).
Author Response
Dear Editor in Chief:
We thank the reviewer for the careful reading of and suggestions for improving our manuscript. Below, we provide a point-by-point response to the reviewer´s comments (our comments appear in italics after reviewer´s comment). The revisions to the original manuscript appear highlighted in red.
Sincerely,
Tiina Mattila, MD
Reviewer 1
Comments and Suggestions for Authors
The review has scientific relevance, however I suggest enriching the study by adding some information such as:
- Search period in the database
We thank reviewer for this comment. This was not a systematic review but a scoping review, and therefore, a fully comprehensive literature search was not made. Original data search in PubMed was made between September 2019 and January 2020 and the search was repeated before this resubmission. After the data search for each chemical was chosen up to 5-6 the most important articles from the recent years. To explain this we added a sentence in Method section. Reviewers 3 and 4 asked about this also. Please see our answers to them also. (Page 3, lines 92-105)
- Maintain a standard of information about each pollutant (chemical structure, sources, transport, reactions with other pollutants, legislation, concentration limit in Europe which can be a range, health effects, ...).
This was an interesting suggestion from the reviewer. We have now systematically explained in the article text the basic chemical background data, source and general health effects for each of the 11 chemical. However, we think that more detailed information about each chemical would have confused the focus which was the association between each reported chemical and asthma.
We added missing data for this part in the text, yet, some of this this data appeared already previously in Table 1.
For this data and as response to reviewer 3 we added new references.
(new references by their first author/ word in reference: Låg et al 2020, Jenerowitz et al 2012, Klingbeil et al 2014, Collins et al 2017, Wisnewski et al 2001, Schneider et al 2012, Zeng et al 2016, Walters et al 2012, Ye et al 2016, Benka-Coker et al 2020, Ye et al 2016, Wittasek et al 2011, Odebeatu et al 2019, Hoppin et al 2013, Bornehag et al 2010, Fischer et al 2017, KEMI 2015, OECD 2015, Eu environment strategy 2021, Smit et al 2015, Jackson-Browne 2020, Averina et al 2019, Fento et al 2020, Kvalem et al 2020, Koh et al 2019, Weidinger et al 2004, Siddique et al 2020, Wang et al 2017, Costa et al 2006. Changes: page 4, lines 144-146, 149-153, 162-163; page 5, lines 180-181, 188-198, 200; page 6, lines 212-215, 237-239; page 7, line 271; page 8, line 305)

Reviewer 2 Report
Authors provide a good summary of concerning environmental chemicals and its association with Asthma.
Author Response
Dear Editor in Chief:
We thank the reviewer for the careful reading of and suggestions for improving our manuscript. Below, we provide a point-by-point response to the reviewer´s comments (our comments appear in italics after reviewer´s comment). The revisions to the original manuscript appear highlighted in red.
Sincerely,
Tiina Mattila, MD
Reviewer 2
Comments and Suggestions for Authors
Authors provide a good summary of concerning environmental chemicals and its association with Asthma.
We thank reviewer for this comment. No changes were made.

Reviewer 3 Report
Strengths:
- This review is short, but highly condensed and to the point. It includes quite a few chemicals/chemical groups that often are not considered related to asthma etiology. So, the review brings new insight and knowledge to the environmental triggers of asthma attack.
- The review was well prepared.
- The three tables are good summary on sources of exposure, routes of exposure, measurement metrics and biomarkers, factors affecting measurement and exposure, populations at risk of exposure and associations with asthma based on current literature. These tables give readers the context and help the understanding.
- The English was good and further editing is mostly not needed. It is ready to publish.
Weaknesses:
- It is not a conventional comprehensive review. It started with chemicals listed in their European Human Biomonitoring Initiative (HBM4EU) as priority substances.
- It does not include all substances related or possibly related to asthma. It is not clear what made them their priority. The focused chemicals or chemical groups were not inclusive.
- The search was limited in terms of database used. The years and keywords used for each were not used. It looks like PRISMA review guidelines were not followed and a diagram to summarize the review was not provided.
- Overall, the references included or cited were limited (total of 40).
Recommendations:
- Although the manuscript has some weaknesses, I think it has merits to be published because it provides new information and look at asthmagens from a different angle.
- Recommend publishing it without further revision.
If the authors are willing to expand the search for more literature using more databases and covering more years, that would be better to increase the depth of review on each chemical group as they relate to the asthma etiology. This way, it will provide more science and information to the readers.
Author Response
Dear Editor in Chief:
We thank the reviewer for the careful reading of and suggestions for improving our manuscript. Below, we provide a point-by-point response to the reviewer´s comments (our comments appear in italics after reviewer´s comment). The revisions to the original manuscript appear highlighted in red.
Sincerely,
Tiina Mattila, MD
Reviewer 3
Comments and Suggestions for Authors
Strengths:
- This review is short, but highly condensed and to the point. It includes quite a few chemicals/chemical groups that often are not considered related to asthma etiology. So, the review brings new insight and knowledge to the environmental triggers of asthma attack.
- The review was well prepared.
- The three tables are good summary on sources of exposure, routes of exposure, measurement metrics and biomarkers, factors affecting measurement and exposure, populations at risk of exposure and associations with asthma based on current literature. These tables give readers the context and help the understanding.
- The English was good and further editing is mostly not needed. It is ready to publish.
We thank reviewer for this positive feedback about our work. No changes were asked to made.
Weaknesses:
- It is not a conventional comprehensive review. It started with chemicals listed in their European Human Biomonitoring Initiative (HBM4EU) as priority substances.
As noticed, this article is not a comprehensive review, yet a scoping review including the most important existing data from the recent years (from 1994 to 2020) about the association between each included chemical and asthma. For a systematic review, there would have been a need for up to 11 reviews (one for each included chemical) considering the current amount of scientific data about this subject. We tried to write balanced article with about the same amount of references and data for each chemical, though in the PubMed there were hundreds of original articles and also systematic reviews written about some of the chemicals while about another there were less than 10 articles. While this was not a systematic review we searched articles only in PubMed and not from other databases. Please see also our response to reviewer 1 and 4 about same kind of questions. We added a reference in Methods section as background information for scoping review (Sucharew et al, 2019), no other changes in article.
- It does not include all substances related or possibly related to asthma. It is not clear what made them their priority. The focused chemicals or chemical groups were not inclusive.
Background for this scoping review included the chemicals prioritized for the HBM4EU project and having an association or a possible association with asthma. HBM4EU is a human biomonitoring initiative in the European scale under Horizon 2020 and it includes 30 European countries. The focus of HBM4EU is to support chemical policies by building knowledge to on the chemical exposure and subsequent impacts on health to the populations in Europe. Prioritized chemicals were chosen by a comprehensive, international group of researchers and experts in structured prioritization process based on the need of knowledge about the health effects of those chemicals. We introduced more details in the methodology part to explain it better in the text. More information found in www.hbm4eu.eu and https://www.hbm4eu.eu/wp-content/uploads/2018/09/Deliverable-4.5-Second-list-of-HBM4EU-priority-substances-and-Chemical-Substance-Group-Leaders-for-2019-2021.pdf. (Page 3, lines 92-98, and previously page 2, lines 82-86)
- The search was limited in terms of database used. The years and keywords used for each were not used. It looks like PRISMA review guidelines were not followed and a diagram to summarize the review was not provided.
This was a scoping review not a systematic review following PRISMA guidelines. Please see the new reference number 29( Sucharew et al, 2019) and our response to you in comment Weaknesses 1 and Reviewer 1 and 4 where changes made in text are explained.
- Overall, the references included or cited were limited (total of 40).
We added references about chemicals background if deficient as response to reviewer 1, comment 2, and relevant new articles about the association between each chemical and asthma. New references are mentioned in our answer to referee 1, comment 2, please see above. (Changes: page 4, lines 134-139, 154-155; page 5, lines 171-175, 178-179, 207-210; page 6, lines 228-231; page 7, lines 249-253, 256-257, 260-262, 271, 286-287; page 8 302-303, 319-322; page 9, lines 335-336)
Recommendations:
- Although the manuscript has some weaknesses, I think it has merits to be published because it provides new information and look at asthmagens from a different angle.
- Recommend publishing it without further revision.
We thank the reviewer 3 for this comment.
- If the authors are willing to expand the search for more literature using more databases and covering more years, that would be better to increase the depth of review on each chemical group as they relate to the asthma etiology. This way, it will provide more science and information to the readers.
Please see our response to your comment above (weaknesses 4) and added references.

Reviewer 4 Report
Scoping Review-the associations between asthma and environmental chemicals
This manuscript is about a review of documents developed for the European Human Biomonitoring Initiative (HBM4EU), so they researched PubMed and developed a summary of the chemicals prioritized in HBM4EU related or possible related to asthma.
The summary of each chemical that the authors described is succinct, but it includes the most relevant information and how affects its risk or not to asthma. They also acknowledge that data is lacking and more research is needed when health effects can be due to combined chemicals.
Comments:
The manuscript does not provide new information related with the knowledge from those chemicals; however, they describe those chemicals directly and succinctly, and covers important aspects about their association with asthma.
Methodology
- The methodology section is very brief, and the authors did not explain in detail how they did the scoping review. They only mention that they used scoping documents developed for all HBM4EU and that they are available in the project’s website, and that they used PubMed including “asthma” and each HBM4EU priority chemical. It will be helpful for readers that they include a more specific methodology of the research question they used for those articles searched in PubMed; what criteria used to identify relevant studies, and study selection.
Tables
The authors include three tables: 1) included exposure routes of some common sources of exposure; 2) matrices commonly used for measuring exposure to substances and factors…., and 3) sensitive population group for each chemical and association with asthma.
- The authors did not include a table of the articles searched with the authors, design of the study, population, exposure, type of health effect, and primary findings.
I believe that such information is needed to provide a better understanding of the review they did and the kind of articles they reviewed.
Discussion
The manuscript does not have a critical discussion of the reviewed articles. There is no comparison or contrast among studies to explain author’s results. The authors do not provide limitations in their study and major or minor finding in the context of the revised literature.
Author Response
Dear Editor in Chief:
We thank the reviewer for the careful reading of and suggestions for improving our manuscript. Below, we provide a point-by-point response to the reviewer´s comments (our comments appear in italics after reviewer´s comment). The revisions to the original manuscript appear highlighted in red.
Sincerely,
Tiina Mattila, MD
Reviewer 4
Comments and Suggestions for Authors
Scoping Review-the associations between asthma and environmental chemicals
This manuscript is about a review of documents developed for the European Human Biomonitoring Initiative (HBM4EU), so they researched PubMed and developed a summary of the chemicals prioritized in HBM4EU related or possible related to asthma.
The summary of each chemical that the authors described is succinct, but it includes the most relevant information and how affects its risk or not to asthma. They also acknowledge that data is lacking and more research is needed when health effects can be due to combined chemicals.
Comments:
The manuscript does not provide new information related with the knowledge from those chemicals; however, they describe those chemicals directly and succinctly, and covers important aspects about their association with asthma.
We thank reviewer for these two comments. No changes were made.
Methodology
- The methodology section is very brief, and the authors did not explain in detail how they did the scoping review. They only mention that they used scoping documents developed for all HBM4EU and that they are available in the project’s website, and that they used PubMed including “asthma” and each HBM4EU priority chemical. It will be helpful for readers that they include a more specific methodology of the research question they used for those articles searched in PubMed; what criteria used to identify relevant studies, and study selection.
We rewrote the methods section for this and reviewer 1 and 3 comments. Please see also our answers for them. (page 3, line 92-105)
Tables
The authors include three tables: 1) included exposure routes of some common sources of exposure; 2) matrices commonly used for measuring exposure to substances and factors…., and 3) sensitive population group for each chemical and association with asthma.
- The authors did not include a table of the articles searched with the authors, design of the study, population, exposure, type of health effect, and primary findings.
I believe that such information is needed to provide a better understanding of the review they did and the kind of articles they reviewed.
Thank you for this suggestion. As we answered previously to reviewer 1 and 3, this article was a scoping review not a systematic review, and therefore no systematic literature search was made. To keep the article in balance our text includes up to 5-6 the most reasonable articles about each included 11 chemical. We think that the Table the reviewer suggested would not have given any more information for this data search. Please see also our responses to reviewer 1 and 3. No changes were made.
Discussion
The manuscript does not have a critical discussion of the reviewed articles. There is no comparison or contrast among studies to explain author’s results. The authors do not provide limitations in their study and major or minor finding in the context of the revised literature.
The function of a scoping review is to make a referred summary about the analyzed subject, in this case chemicals included in HBM4EU project and their association with asthma. Therefore a scoping review should not include a critical discussion, but the Discussion section should more be a conclusion about this referred data including information about what is currently known and what should be studied more in future. No changes were made.
